# Energy Recovery from Polymeric 3D Printing Waste and Olive Pomace Mixtures via Thermal Gasification—Effect of Temperature

**DOI:** 10.3390/polym15030750

**Published:** 2023-02-01

**Authors:** Daniel Díaz-Perete, Manuel Jesús Hermoso-Orzáez, Luís Carmo-Calado, Cristina Martín-Doñate, Julio Terrados-Cepeda

**Affiliations:** 1Department of Graphic Engineering, Design and Projects, University of Jaén, 23071 Jaén, Spain; 2Centre for Advanced Studies in Energy and Environment, University of Jaén, 23071 Jaén, Spain; 3IPPortalegre—Polytechnic Institute of Portalegre, 7300-555 Portalegre, Portugal

**Keywords:** FDM, plastic materials, polymers, thermochemical process, Industry 4.0, 3D print waste recovery, waste to energy

## Abstract

One of the polymeric materials used in the most common 3D printers is poly(ethylene terephthalate) glycol (PETG). It represents, in world terms, around 2.3% of polymeric raw material used in additive manufacturing. However, after processing this material, its properties change irreversibly. A significant amount of waste is produced around the world, and its disposal is usually destined for landfill or incineration, which can generate an important issue due to the high environmental risks. Polymer waste from 3D printing, hereinafter 3DPPW, has a relatively high calorific value and adequate characteristics to be valued in thermochemical processes. Gasification emerges as an innovative and alternative solution for recovering energy from 3DPPW, mixed with residues of lignocellulosic origin, and presents some environmental advantages compared to other types of thermochemical treatments, since the gasification process releases smaller amounts of NOx into the atmosphere, SOx, and CO_2_. In the case of the study, co-gasification of olive pomace (OLB) was carried out with small additions of 3DPPW (10% and 20%) at different temperatures. Comparing the different gasifications (100% OLB, 90% OLB + 10% 3DPPW, 80% OLB + 20% 3DPPW), the best results for the synthesis gas were obtained for the mixture of 10% 3DPPW and 90% olive pomace (OLB), having a lower calorific value of 6.16 MJ/m^3^, synthesis gas yield of 3.19%, and cold gas efficiency of 87.85% for a gasification temperature of 750 °C. In addition, the results demonstrate that the addition of 3DPPW improved the quality of syngas, especially between temperatures of 750 and 850 °C. Including polymeric 3D printing materials in the context of the circular economy and extending their life cycle helps to improve the efficiency of subsequent industrial processes, reducing process costs in general, thanks to the new industrial value acquired by the generated by-products.

## 1. Introduction

Three-dimensional printing was developed in the 1990s, and this technology has grown exponentially around the world, where it is considered a relatively easy-to-use and promising manufacturing approach [1]. Poly(ethylene terephthalate) glycol (PETG) is a thermoplastic polyester commonly used in manufacturing of parts through 3D printing technology. The PET component is what is commonly found in plastic beverage bottles and food products. The G stands for glycol, which adds durability and strength, and contributes to the increment on impact resistance and ability to withstand high temperatures of the compound whose physical and chemical properties are considered suitable for 3D printing [2,3,4]. PETG filament is quickly becoming a popular choice in 3D printing due to its widely industrial applications. It already represents, in world terms, about 2.3% of polymeric raw material used in additive manufacturing [5,6,7]. Despite its advantages, the printing process generates significant amounts of waste, making this material after melting a growing concern in relation to its final destination. In the industrial field, PETG utilization as 3D printing material is especially relevant in the food and dental sectors. PETG is credited by the FDA (Food and Drugs Administration) for its use in food containers and packaging, so is a fantastic option to elaborate prototypes of containers, bottles, glasses, or packages [8]. In the dental industry, it is one of the viable and safe materials in transparent dental aligner production. Aligners are fabricated through additive manufacturing technology and provided as a personalized service to customers. The lifespan of these products is 15–30 days, depending on facultative specific studies developed for the customer. After the lifespan time passes, dental aligners do not have another specific use and become waste [9,10]. Several market studies estimate an important growth of additive manufacturing utilization in the dental sector, positioning PETG as one of the most favored materials for its use in relationship with this technology during this decade [11,12].

According to the Paris Agreement, all countries can include actions on waste management as part of efforts to reduce greenhouse gas emissions, such as, for example, valuing waste as a source of energy, recycling, and reuse [13]. However, and according to the European Union, annual global waste production is expected to jump from 2.01 billion tons in 2016 to 3.40 billion tons over the next 30 years [14]. This point suggests that waste management is not working and that there is a continuous growth in waste production, making us a “disposable society”. It is urgent to carry out an adequate prevention policy to reduce waste production, followed by reuse and recycling. Energy recovery projects can be classified as a type of complementary technology for the recovery of energy from non-recyclable material and, therefore, should not compete with measures for the prevention, reuse, and recycling of waste.

The traditional methods of treating this type of waste are pyrolysis [15], incineration, chemical treatment, and biodegradation [16], but these have some shortcomings. For example, the method of pyrolysis and incineration is energy-intensive, and in addition to this, incineration produces harmful gases such as NOx and SOx [17,18]. Furthermore, the landfill solution, as a destination for 3DPPW, is also not viable [19]. 

Thermal gasification is an effective method of treating and utilizing polymeric waste for energy production [20]. Although the thermochemical treatment of materials of lignocellulosic origin has been well-covered in the literature [21,22,23], the gasification of residues of polymeric origin has been presenting more relevant topics in recent research [24,25,26]. Polymeric residues have a high calorific value, and low moisture and ash content. However, these types of materials may present operational problems during gasification tests if they are introduced without mixing with a raw material of lignocellulosic origin, such as creation bridges, tar formation, and equipment deterioration [27]. The method of mixing two or more raw materials, namely, with biomass, has been attracting attention due to the synergistic interactions between the two, namely, in the quality of the syngas, which promotes an increase in light hydrocarbons [28]. This aspect is highly relevant and advantageous in applications where waste separation becomes complicated, such as in the packaging industry. Several studies were carried out in different types of gasifiers, such as fluidized-bed and fixed-bed reactors. Fluidized-bed-type reactors have advantages in heat transfer, better gas–solid reaction, and good temperature control [29]. Fixed-bed reactors have the advantage of having a simpler design, removal of unconverted material in the reactor bed, and less formation of hydrocarbons [30].

More recently, the co-gasification of polymeric residues and residues of lignocellulosic origin has aroused the interest of many researchers. For example, co-gasification with supercritical water of coal and various types of polymeric waste was studied by Bian et al. [31], and the results indicated that there was a synergistic effect when plastic waste was added. Kaydouh et al. [32] investigated the co-gasification of palm oil fiber waste with high- and low-density polyethylene, and came to the conclusion that the addition of polymeric waste promotes the concentration of hydrogen and carbon monoxide in the syngas. Another author, who carried out the co-gasification of the two types of waste, also reached the same conclusion, that the addition of plastics increases the concentration of hydrogen and carbon monoxide [33].

Thus far, as far as the authors are aware, there are no pilot-scale studies with a mixture of olive pomace residues and a mixture of small fractions of PETG residues originating from 3D printing. Therefore, a comprehensive analysis was carried out on a semi-industrial scale, where several gasification tests were carried out, with different mixtures and at different temperatures, which was the main objective and novelty of this study. Gasification performance was studied in terms of the lower calorific value of the syngas and, consecutively, its composition, syngas yield, and energy efficiency, demonstrating that the addition of polymeric residues has some advantages for the studied parameters, namely, at high temperatures. This study may allow the optimization of decentralized units on a pilot scale, diversifying the energy matrix and suitable for local conditions.

## 2. Materials and Methods

Waste generated through 3D print polymer residues after mechanical tests can be energy recovered (waste-to-energy) through thermochemical processes, in this case used as fuel in combination with olive pomace biomass, henceforth olive pomace, through the thermal gasification process. Polymer residue characteristics seem, a priori, appropriate in combination with lignocellulosic materials to act as fuel in a thermal gasification process due to the plastic waste present in their composition, high content of hydrogen, and low concentration of oxygen [34]. Otherwise, until its definitive solution, in the form of waste recycling or energy recovery, plastic waste is still deposited in landfill, causing environmental problems.

To elaborate, the experimental gasification test olive pomace biomass (OLB) and PETG 3D printing residues were used in different proportions of mixture for fuel preparation. The experimental tests were conducted at different temperatures too, in order to perceive the influence of and change in the rest of the gasification-related parameters in concordance with temperature changes. Composition of fuels studied is 10/90 and 20/80 of 3DPPW/OLB, respectively. The chosen fuel compositions respond to the fact that experimental tests conducted with 20% polymer material composition are not functional, due to polymeric material viscosity when melted causing gasification reactor obstruction [35]. A 100% OLB gasification was also conducted, using the same boundary conditions of temperature and operational conditions, in order to have a comparison and a measure of how much results change in concordance with the 3DPPW proportion added to biomass fuel. Experimental test temperatures defined in this manuscript are 650 °C, 750 °C, and 850 °C. Monitored parameters in these experimental gasification tests were: raw material consumption, oxidant agent flow, efficiency ratio, quantity and quality of produced syngas, and tar and char quantity produced.

### 2.1. Raw Material Analysis

#### 2.1.1. Ultimate Analysis

Elemental analysis enables knowledge of the chemical composition of raw materials. Nitrogen (N), carbon (C), hydrogen (H), sulfur (S), and oxygen (O) are the interesting elements included, for which quantities were defined using a Thermo Fisher Scientific Flash 2000 CHNS-O device.

#### 2.1.2. Proximate Analysis

Thermogravimetric analysis was used to establish moisture, volatile matter, fixed carbon, and ashes of raw material samples. A PerkinElmer STA 6000 device was used to determine the abovementioned parameters, with the heat rate fixed at 20 °C/min. Finally, raw material composition was established through the thermogravimetric curve (mass variation of sample versus temperature), considering tipping points of the mass derivative as a function of time.

#### 2.1.3. Higher Heating Value

Higher heating values (*HHV*s) of fuels were calculated using an IKA C 2000 device, through an adiabatic complete combustion of samples.

#### 2.1.4. X-ray Fluorescence

The X-ray fluorescence (XRF) technique was used to detect chlorine concentration in samples, to verify the viability of the process, avoiding corrosion in equipment produced by hydrochloric acid after chlorine transformation during thermochemical processes. The XRF analysis device used was the Thermo Scientific Niton XL 3T GoldD +.

### 2.2. Gasification Tests

Gasification tests were conducted using a downdraft fixed-bed gasifier reactor, as the scheme presented in Figure 1 shows. The gasification system is composed of one fuel storage silo, with a heat exchanger that uses hot gas stream recirculation produced in the reactor. Fuel deposition is performed from the top of the reactor, and likewise with the oxidant agent. In this case, the oxidant agent is air, which in contact with reactor walls is preheated.

At the bottom of the gasifier reactor, there is a mechanical shaker and a receptor whose objectives are cleaning and storing of char and ash particles and their separation from produced syngas. Syngas leaves the reactor at temperatures close to 500 degrees Celsius, passing by a cyclone filter in order to remove less-dense particles that could be content in syngas flow. After that, syngas is filtered by biomass filters formed by different granulometry biomasses, respectively. In this device, heavy-hydrocarbon and condensable-material retention happen. Finally, clean syngas is analyzed, and afterwards, it can be burned or injected into an internal combustion engine.

Experimental gasification tests were elaborated in terms of co-gasification with OLB and 3DPPW. OLB material is very abundant in the region of Jaén and presents favorable characteristics for thermochemical processes, due to the low content of char and ash and high content of carbon. On the other hand, the fuel component 3DPPW base is formed by PETG obtained from a previously conducted study about PETG characterization manufactured with fused deposition modeling (FDM) technology for architectural applications developed by Mercado-Colmenero et al. [36]. In the case of the comparison tests, 100% OLB fuel gasification was made at defined temperatures, 650 °C, 750 °C, and 850 °C, to study the changes in results with 3DPPW addition in fuel in concordance with the mentioned temperature variations.

The duration of gasification tests was close to 300 min. During this time, gasification parameters such us temperature, pressure, oxidant agent flow rate, fuel consumption quantity and quality, and quantity of obtained syngas were controlled. Syngas test tubes or specimens were collected at the moment the gasification system was stabilized in terms of reactor temperature. By-products obtained after the test, namely, chars and tars, were collected and counted too.

### 2.3. Produced Syngas Analysis

Syngas specimens obtained during tests were analyzed using a Varian 450-GC gas chromatograph, and a thermal conductivity detector (TCD) is also included in this device. It was used for the identification and quantification of gas compounds that form the gas, CO, CO_2_, H_2_, CH_4_, and hydrocarbons presented in syngas. 

### 2.4. Theoretical Parameters

#### 2.4.1. Efficiency Ratio

The efficiency ratio (*ER*) parameter indicates quantitatively whether a fuel mixture is rich, poor, or stoichiometric in terms of energy [37]. This parameter is defined by the following Equation (1):(1)$$ER=\frac{\left(A/F\right)}{\left(A/F\right)stoichiometric}$$
where *ER* is the efficiency ratio, and the (*A/F*) ratio, where *A* represents air and *F* represents fuel, is the relation between these elements in the experimental test boundary conditions. The (*A/F*) *_stoichiometric_* ratio defines the relation between air and fuel in stoichiometric conditions. Therefore, for high-energetic-value mixtures or rich mixtures, *ER* is higher than 1. In the case of poor mixtures, *ER* is lower than 1, and for stoichiometric mixture *ER* has the value 1.

The required oxidant agent quantity to consume all the fuel in order to produce thermal energy is defined by the stoichiometric or theoretical quantity for a combustion system. The gasification process works with a lower oxidant agent level than stoichiometric to obtain chemical energy in the form of syngas. The stoichiometric quantity of a process is calculated through elemental analysis of fuel.

The efficiency ratio parameter defines the performance of gasification. It takes values between 0.2 and 0.4. On the contrary, in the pyrolysis process, *ER* takes a value of 0.

Too-high or too-low values of the efficiency ratio present problems during the gasification process. For lower values than 0.25, biochar cannot be converted totally to syngas. Furthermore, under this boundary condition, high levels of tars are produced, so equipment suffers corrosion or blockage. For high values of *ER*, part of the resultant syngas is burned. This supposes its energetic quality reduction. Furthermore, the reactor temperature elevation causes the formation of slag, and equipment degradation [38].

The gas quality obtained through gasification process has a direct relationship with the efficiency ratio. It should present values so much below 1 as to guarantee that fuel converts into gas and does not burn. In fixed-bed gasification methodology, the *ER* value is fixed at 0.25 [39].

#### 2.4.2. Cold Gas Efficiency (*CGE*) and Syngas Yield

The cold gas efficiency parameter is used to determine the gasification process efficiency. It is expressed through the produced syngas cold energy and the energy of fuel used in the gasification process ratio. In this paper, *CGE* calculations are based on the lower heating value (*LHV*). In Equation (2), the *LHV* of fuel is calculated [40].
(2)$$LHV\;\left(\frac{MJ}{kg}\right)=HHV-0.212\times H_{2}-0.0245\times \mathrm{Mo}-0.008\times O_{2}$$
where *LHV* is the lower heating value, *HHV* is the higher heating value obtained for fuel from the calorimetry test, H_2_ denotes the hydrogen percentage obtained from the elemental analysis test, Mo is the fuel percentage of moisture obtained from thermogravimetric analysis, and O_2_ marks the percentage of oxygen obtained from elemental analysis.

*CGE* determination is performed through Equation (3):(3)$$CGE\;\left(\%\right)=\frac{ṁ\;syngas\times LHV\;syngas}{ṁ\;fuel\times LHVfuel}\times 100$$
where ṁ *syngas* is the flow mass rate from the gaseous product (kg/s), ṁ *fuel* is the flow mass rate from fuel, *LHV syngas* is the lower heating value of syngas, and *LHV fuel* is the lower heating value of fuel.

Syngas yield [41] is defined through Equation (4):(4)$$\eta syngas\;\left(\frac{m3}{\mathrm{kg}}\right)=\frac{ṁ\;volumetric\;syngas}{ṁ\;massic\;biomass}$$
where *ηsyngas* means the syngas yield (m^3^/kg), ṁ *volumetric syngas* is the flow volume rate of syngas (m^3^/h), and ṁ *massic biomass* is the flow mass rate from gasificated fuel (kg/h).

## 3. Results and Discussion

Table 1 shows obtained results for proximate analysis, ultimate analysis, and higher-heating-value analysis from the studied fuels.

Carbon is one of the most important components in terms of thermochemical processes because *HHV* increases depending on biomass carbon content, approximately between 44.1% and 75.5% for cellulose-based biomass [42]. In the case of OLB, carbon concentration value is 52.9%. For 3DPPW, the value reaches 89.9%. In terms of hydrogen, OLB and 3DPPW present concentration values of 7.7% and 13.4%, respectively. Fuel nitrogen content in gasification favors the formation of NOx [43]. The oxidant agent used in the reactor presents a level lower than stoichiometric, so in the gasification, for sub-900 °C temperatures CO and CO_2_ are principally formed and NOx is released in a residual way [44]. The studied fuels present low hydrogen concentration, lower than 1.7%. In terms of sulfur, 3DPPW fuel presents 1.9% concentration. This level can generate some trouble in the form of gaseous emissions. Oxygen concentration presented is 33.2% in the case of OLB and 2.8% for 3DPPW fuel. Oxygen existence presents negative influence related to the *HHV* of fuels. Therefore, sometimes fuels are treated to decrease oxygen levels in the fuels to reach higher values of *HHV*.

Humidity values are 9.4% for OLB and 0.2% in the case of 3DPPW. In combustion processes, this parameter should not be higher to 30% due to thermal losses. However, in gasification processes, the humidity parameter assumes higher values because downdraft reactors use part of the generated reactor energy to dry off the fuel before beginning the process.

The 3DPPW fuel presents volatile matter values of 79.3%. This high number indicates that it is not necessarily high energetic values that start the thermochemical reactions. This means that 3DPPW can be considered a good fuel for a low-temperature gasification system in comparison with another fuel with higher levels of fixed carbon. OLB volatile matter concentration is 65.7%, a normal value for a forest biomass. Fuels with moderate quantities of volatile matter are good in a gasification process as tar formation is reduced. This reduction means less trouble in the maintenance of gasification devices.

High fixed-carbon content presents benefits in thermochemical processes because the energetic density of fuel is increased due to the increment of this parameter. Additionally, the gasification reactor temperature increment is defined by fixed-carbon level too. Consequently, tar thermal cracking is increased too due to this value during the devolatilization process. Raw materials contain 19.9% and 20.5% for 3DPPW and OLB, respectively [45].

Fuel ash contents are very different. In the case of OLB, this value is close to 4.4%, and 0.6% for 3DPPW. The ash percentage content presented is acceptable for a thermochemical process. This parameter presents trouble when the value is higher than 7%, due to slag formation at temperatures higher than 1000 °C. Additionally, the ash content parameter reduces the *HHV* value.

The obtained *HHV* value for 3DDPW is 40.7 MJ/kg. Due to the high value of *HHV*, this fuel cannot be introduced isolated in the reactor because the high temperatures reached during the process can damage the gasification device. The *HHV* value for OLB biomass is 20.3 MJ/kg, which is a common value for a forest dry biomass. Consequently, a mix based on these two raw materials is quite good for gasification as long as the addition limit of 3DPPW is not reached.

### 3.1. Gasification Results

Table 2 shows obtained results for experimental tests of gasification and co-gasification, attending next the ratio of raw materials (3DPPW/OLB) 0/100, 10/90, and 20/80. In the table, production values, composition, *LHV* of syngas, char production level, and tar production level are presented as functions of the selected fuel.

Specimens were collected at 120, 180, 240, and 300 min under gasification test criteria. Temperature sensors are located on the external wall of the reactor. Due to this, oxidation and reduction temperatures seem lower in comparison with standard gasification processes. Table 2 collected data refer to the mean results obtained for four specimens during gasification tests, respecting selected mixtures. Table 1 shows obtained results for proximate analysis, ultimate analysis, and higher-heating-value analysis from studied fuels.

### 3.2. Gasification Temperature Effect on the Studied Mixtures

Gasification temperature is an important parameter which is directly related to the final composition of obtained syngas and its properties. Gasification reactions are principally endothermic, so are enhanced by increased temperature [46]. Figure 2a–d show the gasification temperature’s influence over the obtained syngas composition, with CO_2_, H_2_, C_n_H_m_, and CO_2_ production levels for the different mixes of used fuels.

In a perfect gasification system, and according to the Le Chatelier principle, in endothermic reactions the temperature value marks obtained products. In a gasification process, an increase in temperature means an increase in CO and H_2_ levels of obtained syngas and a decrease in temperature means higher levels of CO_2_ and C_n_H_m_ [42,43,44,45,46,47]. In 10% and 20% 3DPPW mixtures, this phenomenon is observed for temperatures of 650 °C and 750 °C. For every mixture, we consider extreme tests those at 850 °C of temperature, because tests were performed at the limit of good performance of the gasification device. 

Generally, a temperature increase from 650 °C to 850 °C implies a hydrogen concentration rise in syngas from 13.01% to 15.75% in the case of 3DDPW 10% mixture, and 12.59% to 13.67% for 3DPPW 20% mixture. OLB syngas decreases its hydrogen concentration from 14.19% to 13.30%. CO concentration presents increases too, from 19.34% to 19.44% and 18.39% to 20.69% for 10% and 20% 3DPPW mixtures. In the case of 100% OLB fuel, CO concentration decreases from 20.92% to 18.31% due to gasification temperature increase. Gas increases, hydrogen, and CO, are produced essentially due to Boudouard endothermic reactions (C + CO_2_ ↔ 2CO) and thermal cracking (C_n_H_m_ + CO_2_ ↔ 2CO + 2H_2_) [48]. For the temperature rise for 3DDPW mixtures, the abovementioned reactions are dominant and H_2_ and CO levels increase. OLB follows the opposite behavior because CO concentration becomes lower due to temperature increase. This means that, for OLB gasification, Gibbs free energy, CO production responsibility, through the Boudouard reaction, is minor, so CO concentration in syngas tends to remain stable [49]. On the other hand, hydrocarbon and methane level production in the produced syngas show a constant trend with 3DPPW mixture fuels, decreasing its concentration for 750 °C degrees and increasing again at 850 °C. In the case of OLB, methane concentration increases for 750 °C and decreases for 850 °C. Contained syngas hydrocarbons are produced due to tar cracking. This explains the constant hydrocarbon composition level for all temperatures tested for the three gasification fuels used in this study. 

For 100% OLB, 90% OLB and 10% 3DPPW, and 80% OLB and 20% 3DPPW fuel compositions, obtained syngas characteristics were analyzed in terms of *LHV*, *CGE*, and *ɳsyngas* for different temperatures of gasification, as shown in Figure 3a–c.

A gasification temperature increase from 650 °C to 750 °C improves syngas performance for every performed test, as, under these temperature conditions, thermal cracking reactions and Boudouard reactions happen more often. The 90% OLB, 10% 3DPPW mixture fuel presents the same behavior at 850 °C too [50]. However, a gasification temperature increase to 850 degrees Celsius produces a little decrease in syngas performance for the 100% OLB and the 80% OLB, 20% 3DPPW mixtures. This situation is related to Gibbs free energy and the Boudouard reaction, because CO concentration tends to remain stable and, furthermore, there is a higher creation of by-products, chars and tars, meaning it cannot complete its transformation into syngas during the gasification process at 850 °C.

When temperature increases, tests present an increase in *LHV*, for which maximum values reach 6.2 MJ/Nm^3^ for 90% OLB, 10% 3DPPW mixture at 850 °C and, at 750 °C, 5.84 MJ/Nm^3^ in the case of the 100% OLB and 80% OLB, 20% 3DPPW fuels. In general terms, at very high temperatures, greater quantities of CO_2_ and H_2_ are produced, and, consequently, *LHV* increases [51].

*LHV* increase and higher syngas performance mean a *CGE* improvement for all mixtures used as fuels. The maximum percentage obtained is presented by 90% OLB, 10% 3DPPW mix at 850 °C of gasification temperature, and this value is 87.85%. For the 100% OLB and 80% OLB, 20% 3DPPW fuels, the highest values of *CGE* are 80.60% and 73.81%, respectively, at 750 degrees Celsius of gasification temperature.

#### Efficiency Ratio (*ER*)

The oxidant agent of the gasification process is air. Therefore, this combustion process presents a relatively lower heating power compared to other similar processes, due to high N_2_ concentration levels. On the other hand, air as the oxidant agent reduces the costs of the gasification process in comparison with other combustion processes [52]. In gasification, O_2_ concentration in chemical reactions is difficult to determine because N_2_ dissolution must be controlled. These effects produce minor variations in fuel admission which alter syngas composition. Due to this, efficiency ratio analysis became important.

Efficiency ratio obtained values are situated between 0.20 and 0.25. These values present similarity with the scientific literature, which registers values between 0.20 and 0.40 for similar gasification processes [53].

*ER* influence on molar fraction of gases which form the produced syngas are presented in Figure 4a–d.

Initial analysis shows that an *ER* increase affects N_2_ content. When *ER* increase, when air is used as the oxidant agent, the obtained syngas has a higher level of dissolution in nitrogen, so the resultant syngas has minor heating power. The H_2_ and CO increase is explained by reaction temperature increment. A higher oxidant agent flow rate favors exothermal reactions and, consequently, thermal cracking reactions produce higher concentrations of CO and H_2_. *ER* has a negative effect on C_n_H_m_ content. Thermal cracking reactions are increased due to higher temperatures and this causes a decomposition of methane [54]. H_2_ and CO concentration promotion do not favor an increase in combustion reactions. Therefore, CO_2_ formation is not notorious.

*HHV* tends to decrease with *ER* growth. The *HHV* of a raw material is related to quantities of carbon (C) and H_2_ in its composition. Higher values in the abovementioned components allow more production of hydrogen and CO. However, though this is the general trend, other raw material properties can modify this behavior in the case of the gasification process.

### 3.3. Syngas Quality

Syngas quality indicates its utility and destiny of use. There are two ways to define the quality of syngas. The first is based on CH_4_/H_2_ ratio (methane/hydrogen ratio). This measure tends to be used for domestic purposes. For industrial purposes, it is preferable to analyze H_2_/CO ratio (hydrogen/carbon monoxide ratio) [55]. 

In addition, there are other factors or parameters which define syngas quality, for example, carbon conversion parameter (CC), cold gas efficiency (*CGE*), and tar content of syngas.

CH_4_/H_2_ ratio has been studied for different mixtures and temperatures of gasification. Operational condition influence is shown in Figure 5.

Figure 5 shows that CH_4_/H_2_ ratio tends to decrease when temperature increases, except for 100% OLB fuel. This means an increase in gasification temperature tend to increase H_2_ concentration. Additionally, as a result of thermal cracking reactions, with a gasification temperature increase tends to decrease CH_4_ concentration. At 750 °C gasification temperature or higher, results of gasification raw materials become more similar. This fact is explained by the constant decrease in CH_4_ concentration for this temperature range. H_2_ concentration is less pronounced too [34].

Raw materials with polymeric content present a higher CH_4_/H_2_ ratio, principally at lower temperatures. This fact shows the evidence that the gasification process is very much influenced by raw material composition. Elemental analysis has demonstrated that polymeric raw materials have higher carbon and hydrogen content, which favor more combustible gas formation. Higher hydrocarbon content, for example CH_4_, is related to a higher volatile matter content. This explains the different behavior of raw materials at low gasification temperatures.

Some articles in the scientific literature discuss that H_2_/CO ratio has the greatest impact to define the final use for produced syngas [56]. An elevated H_2_/CO ratio allows use of syngas in solid oxide fuel cells (SOFC), which needs a cleaner gas for its operation [57]. Medium values of H_2_/CO ratio become appropriate for Fischer–Tropsch (FT) processes to obtain liquid fuels. In the case of lower ratio values, functionality continues with FT processes but acting as a catalyst [58].

Figure 6 shows the gasification temperature effect on H_2_/CO ratio.

Experimental tests show that, at higher temperatures for all gasifications made to all mixtures, hydrogen is stabilized, and carbon monoxide too. Lower gasification temperatures produce H_2_/CO ratio increases. This phenomenon is explained by Gibbs free energy increase and the Boudouard reaction, which produce CO. Therefore, thermal cracking reactions presence become lower. In these temperature conditions, the first fact dominates over the second producing a decrease in H_2_/CO ratio, agreeing in these terms with J. Xiao et al. [59].

In order to transform syngas into methanol or naphtha, the abovementioned ratio value should be higher than 1.70. It is possible to see in the experimental tests realized that obtained results still present distant values. This is explained by the type of gasification system. The atmospheric type reactor uses air as oxidant agent. However, in general, water steam injection as oxidant agent present better values closer to 1.70 [55].

#### 3.3.1. Syngas Yield

Synthesis gas (SG) or syngas yield is defined as the total number of moles produced per unit of feedstock.

At elevated temperatures, it is shown (see Figure 7) that more syngas is produced due to biochar decomposition and volatile matter thermal cracking. As shown in Table 2, there is a reduction trend in tar and char quantities while syngas level increases, that is, reduces to molecules with lower molecular weight [60]. Additionally, syngas *LHV* increases with temperature increase, due to higher concentration of H_2_, CO, and CH_4_. Finally, 3DPPW as the additive in fuels has a significant impact on tar and char performance, agreeing with Ali Abedi et al. [41].

#### 3.3.2. Cold Gas Efficiency (*CGE*)

*CGE* is defined as the chemical energy contained in a gaseous product in connection with contained energy in the initial fuel. Figure 8 shows the relationship between *CGE* and gasification temperature parameters, in concordance with the rest of the operational used parameters.

Gasification temperature presents positive influence on *CGE* due to endothermic reaction increase. These reactions are supported by temperature and syngas yield increases, being both parameters principally important in *CGE* value [61,62].

As expected, OLB fuel *CGE* obtained is minor due to the lower heating power in comparison with polymeric-mixture-composed fuels. This fact applies to lower syngas yield for OLB fuel too [63].

### 3.4. Correlation between Gasification Parameters

Experimental data were statistically analyzed to create main regression and correlation equations to determine the more significant parameters in terms of hydrogen production in syngas. To elaborate this analysis, an exergetic optimization software model has been developed.

To understand variables’ influence on produced syngas, correlation between the main operational parameters was studied. A Pearson correlation model measures force and direction between two continuous variables related to another parameter. In this case, the chosen variables are temperature and *ER*, and the response parameter is hydrogen concentration in syngas.

Experimental tests made for 100% OLB fuel, as shown in Figure 9, present a negative correlation between *ER* and hydrogen concentration of syngas.

It is shown that fuel composed by raw material 100% formed by olive pomace, strongly depends on the *ER* variable. The oxidant agent, in the case of the gasification system, is air, which enters in the reactor as a negative factor of influence for hydrogen concentration. The temperature parameter presents a small correlation, but it is possible to conclude that a temperature increase favors hydrogen concentration in syngas. 

Experimental tests made for 10% 3DPPW and 90% OLB mixture fuel present different correlation behaviors, as shown in Figure 10.

For this fuel, Pearson correlation indicates a positive correlation between temperature and hydrogen concentration, which means that hydrogen concentration is strongly related to reactor temperature increase. Figure 10 shows that *ER* parameters have a negative influence on hydrogen concentration of syngas. 

The 20% 3DPPW 80% OLB mixture tends to similar behavior to the abovementioned mixture. Temperature influence on hydrogen concentration is elevated, as shown in Figure 11.

The figure above shows hydrogen content in the produced syngas for this mixture which forms the fuel. It is shown that hydrogen concentration strongly depends on gasification temperature, and *ER* parameters offers lower dependence. In conclusion, polymer addition into olive pomace biomass (OLB) forms compounds which, in terms of hydrogen concentration on syngas, are strongly favored by elevate gasification temperatures. Furthermore, Pearson correlation results obtained for both parameters, *ER* and temperature, indicate that raw material composition presents great influence on the gasification process and, consequently, in the produced syngas composition. 

### 3.5. Numerical Model

In consideration of previous sections of this study, gasification parameter influence on final syngas composition is very important. 

To obtain as high a quality of syngas as possible and to understand gasification parameter influence and its acting mechanisms, it is necessary to elaborate experimental gasification tests and focus on the abovementioned parameter behavior details. Without a systematic path to study this process, it could become slow and too expensive. Additionally, computational numerical optimization models can be used, the DOE (design of experiments) technique, for example, in order to obtain virtual simulations of the gasification process and avoid some costs and, additionally, establish optimal operative conditions in concordance with the desired results [64]. 

An empirical mathematical model has as its objective of study the concentration of hydrogen in syngas. This is based on gasification temperature and efficiency ratio (*ER*). This model is defined in Equation (5).
(5)Y= β0+ β1X1+β2X2…+ ε
where *Y* represents hydrogen concentration, *β* is the intersection, *X*1 is the temperature predictor, the *X*2 predictor is related to *ER*, and *ε* represents random error. 

A strong regression estimates expected values and predicts impacts of future alterations. Simulated test results keep no relation with the experimental test made. Simulated results are obtained through the described numerical method under the boundary conditions of the experimental gasifications [65]. 

Figure 12 (100% OLB), Figure 13 (10% 3DPPW), and Figure 14 (20% 3DPPW) show exergetic efficiency for hydrogen concentration of produced syngas relative to temperature and *ER* parameters for 100% OLB and 3DPPW mixture fuels. 

The figures above, related to simulations made, show similar behavior to that previously described in experimental gasification tests, pointing to the importance of raw material and fuel composition in terms of obtained syngas characteristics through gasification processes. The 100% OLB fuel tends to increase its hydrogen concentration, establishing a strong dependency between hydrogen percentage and the *ER* parameter. Hydrogen exergy tends to decrease when *ER* increases. In this fuel, the temperature parameter presents low influence in terms of exergy. In the case of 3DPPW fuel mixtures, the similarity between them was described in previous sections of this manuscript. In general, hydrogen efficiency improves according with the temperature parameter, because exergy provided by reactions favors thermal cracking reactions. However, steam addition needs additional exergy to work [66].

Agro-industry waste and 3DDPW compounds present very different behaviors in terms of maximum hydrogen efficiency. The 3DPPW mixtures have higher molar hydrogen composition and present higher-yield syngas. Meanwhile, OLB fuel presents lower values of exergy. Finally, 3DDPW mixtures present a bigger C_x_H_y_ fraction in their composition. This fact involves *CGE* values obtained being significantly higher [66].

## 4. Conclusions

The main conclusions that can be drawn from the olive pomace biomass tests with additive manufacturing waste formed by poly(ethylene terephthalate) glycol are the following:

Firstly, co-gasification test analysis indicates that the objective of energy valorization of 3D printing waste formed by PETG is possible following this path, and results verify and hold this fact. Polymer addition in biomass olive pomace fuel composition forms a composed fuel which favors the production of high-hydrogen-percentage syngas at elevated temperatures, in terms of gasification. In addition, Pearson’s correlation obtained for efficiency ratio and temperature, indicates that, for compositions of up to 20% 3DPPW (PETG) in the fuel, high-quality syngas is obtained, reaching values of around 15% H_2_ concentration in its composition.

For mixed fuel, it is very interesting the reduction in H_2_/CO ratio, this fact happens at high temperatures, higher than 750 °C. The *LHV* of syngas increased according to temperature increase too, due to bigger concentrations of H_2_, CH_4_, and CO. As expected, 100% OLB (olive pomace biomass) fuel *CGE* (cold gas efficiency) is lower than the mixed-fuels case. This is explained by the fact that OLB fuel *HHV* is lower than composed fuels. Consequently, obtained syngas from 100% OLB fuel presents lower yield too. 

Finally, in reference to H_2_ percentage contained in syngas, mixed fuels with 3DPPW content present more uniform and less dispersed behaviors in every temperature range. Due to this, H_2_ content of produced syngas is linearly stabilized. 

In summary, it can be concluded that thermal gasification is a good path to obtain the energy recovery of PETG waste from 3D printing. This process allows mixture of around 20% waste material as the additive for olive pomace biomass fuels obtaining high-energetic syngas with an elevated proportion of H_2_ content and without altering the emissions of polluting gases in the process.

In consideration of new research lines, gasification of 3D printing polymer waste generated through additive manufacturing processes, a greater cleanliness of the syngas obtained should be attempted, the achievement of syngas with higher yields, orienting towards types of syngas with a higher fraction of hydrogen, for its reuse and storage as solid oxide fuel cells.

## Figures and Tables

**Figure 1 polymers-15-00750-f001:**
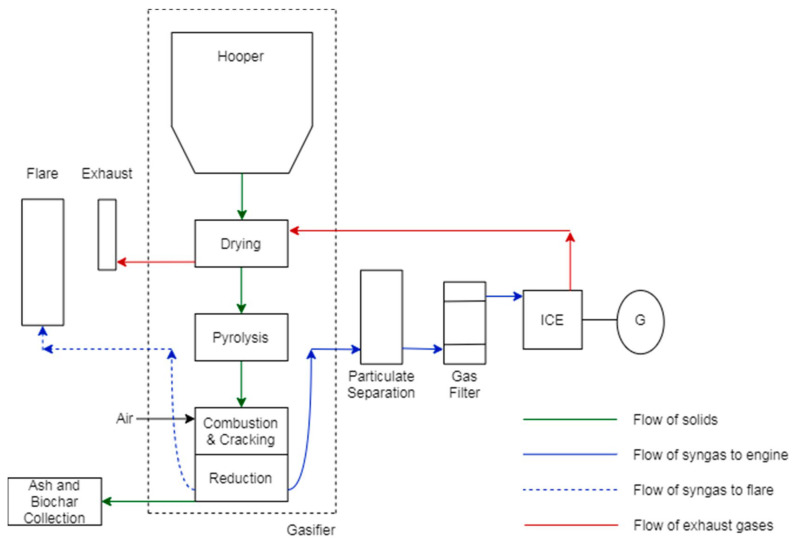
Gasification device scheme.

**Figure 2 polymers-15-00750-f002:**
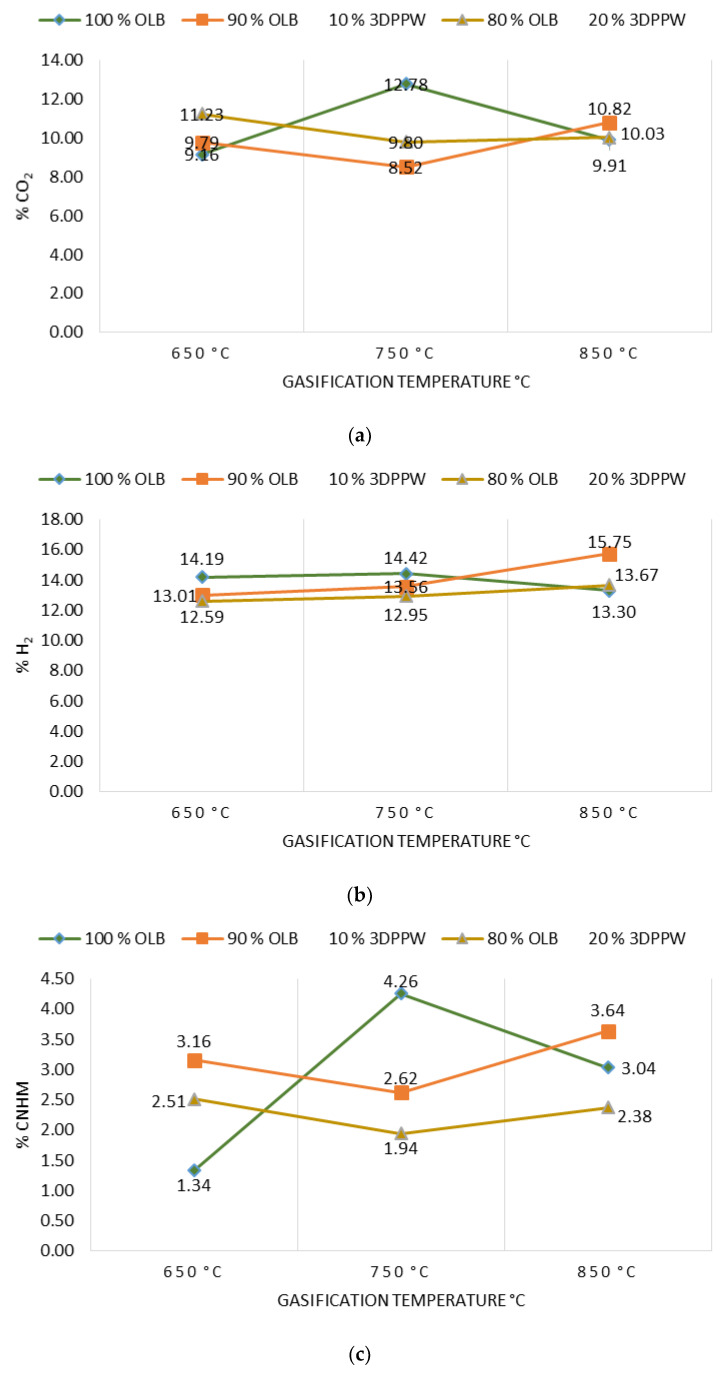
(**a**) Effect of temperature on % CO_2_. (**b**) Effect of temperature on % H_2_. (**c**) Effect of temperature on % of hydrocarbons. (**d**) Effect of temperature on % CO.

**Figure 3 polymers-15-00750-f003:**
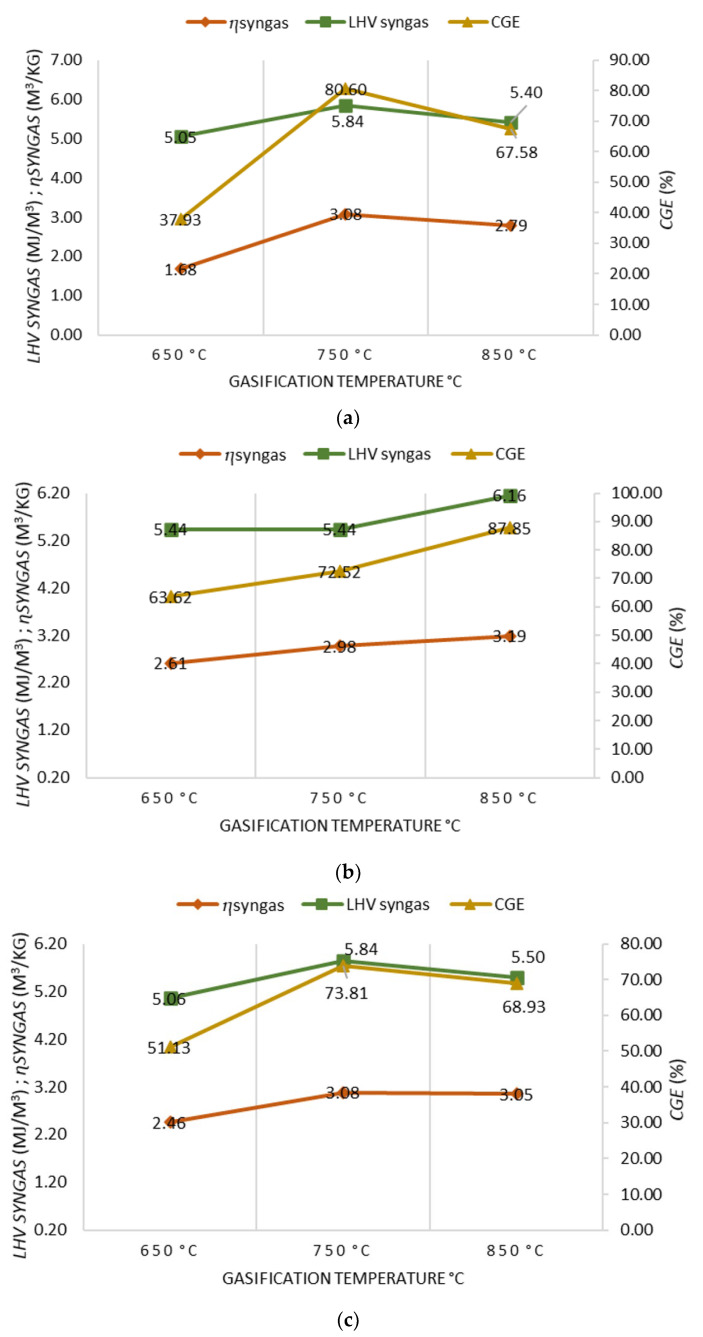
(**a**) Effect of gasification temperature on gasification performance for 100% OLB. (**b**) Effect of gasification temperature on gasification performance for 90% OLB, 10% 3DPPW. (**c**) Effect of gasification temperature on gasification performances for 80% OLB, 20% 3DPPW.

**Figure 4 polymers-15-00750-f004:**
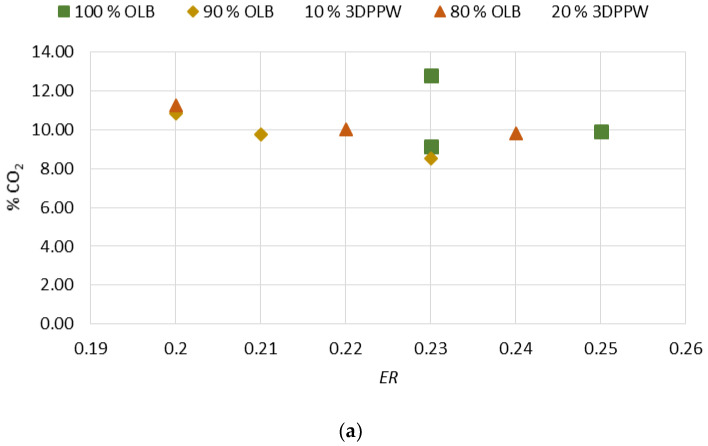
(**a**) Effect of *ER* on % CO_2_. (**b**) Effect of *ER* on % H_2_. (**c**) Effect of *ER* on % hydrocarbons. (**d**) Effect of *ER* on % CO.

**Figure 5 polymers-15-00750-f005:**
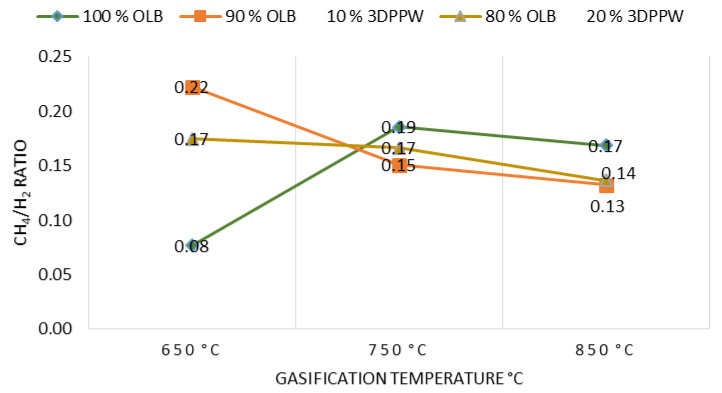
CH_4_/H_2_ ratio as a function of gasification temperature for the different mixtures.

**Figure 6 polymers-15-00750-f006:**
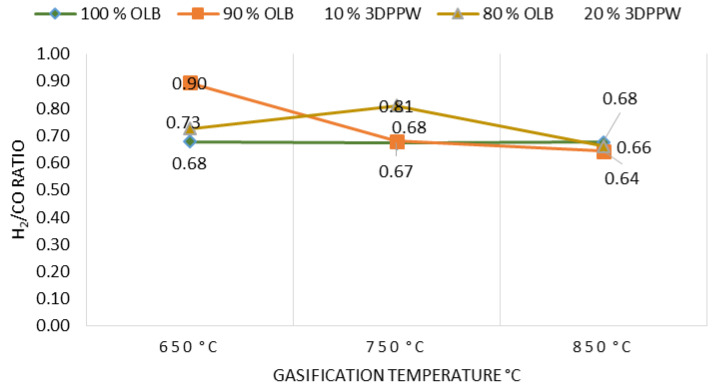
H_2_/CO ratio as a function of gasification temperature for the different mixtures.

**Figure 7 polymers-15-00750-f007:**
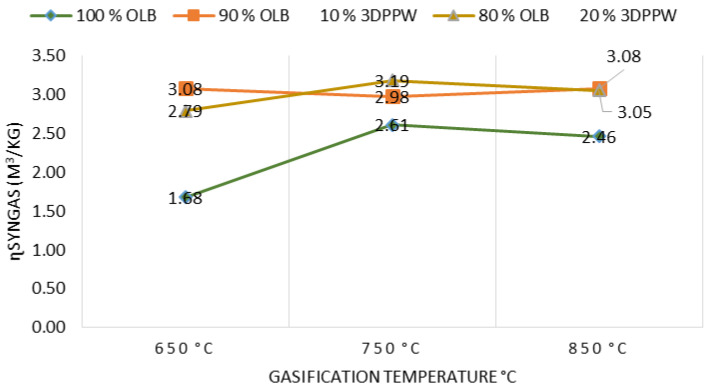
Effect of temperature on syngas yield.

**Figure 8 polymers-15-00750-f008:**
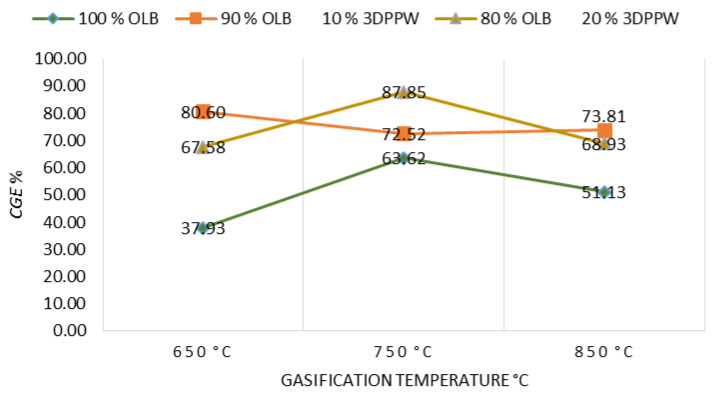
*CGE* as a function of gasification temperature for the different mixtures.

**Figure 9 polymers-15-00750-f009:**
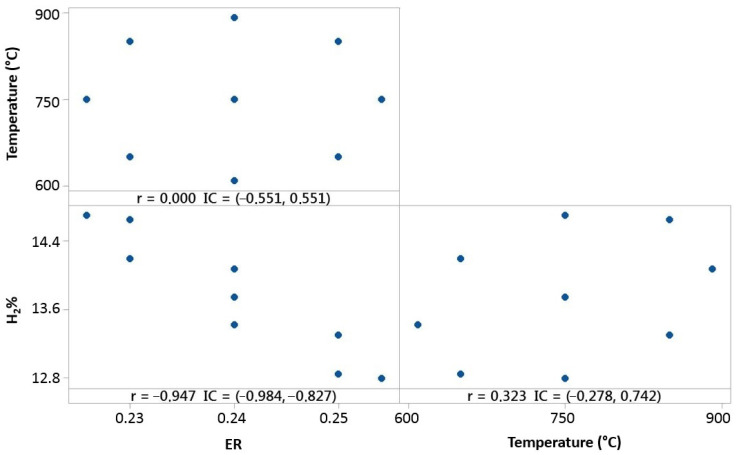
Temperature, *ER*, and hydrogen distribution matrix for Pearson’s correlation for 100% OLB.

**Figure 10 polymers-15-00750-f010:**
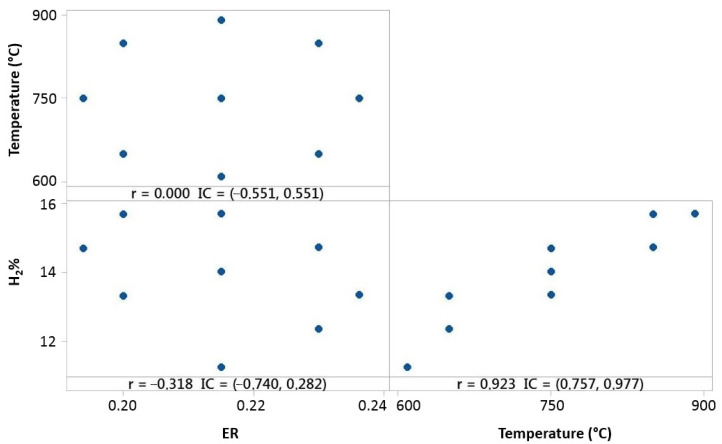
Temperature, *ER*, and hydrogen distribution matrix for Pearson’s correlation for the 90% 3DPPW mixture.

**Figure 11 polymers-15-00750-f011:**
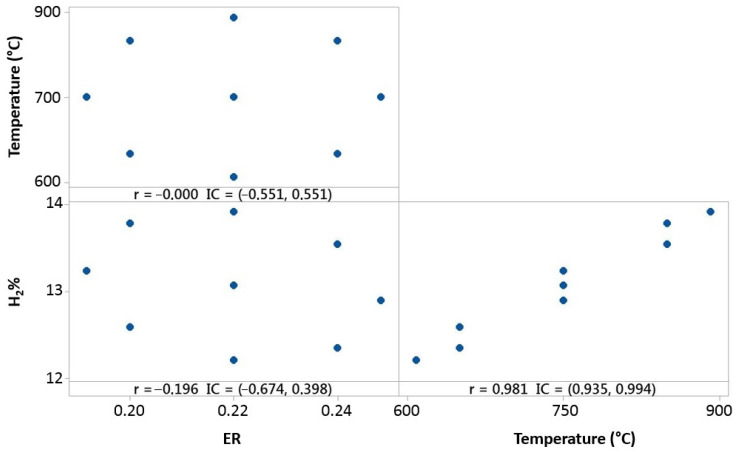
Temperature, *ER*, and hydrogen distribution matrix for Pearson’s correlation for the 80% 3DPPW mixture.

**Figure 12 polymers-15-00750-f012:**
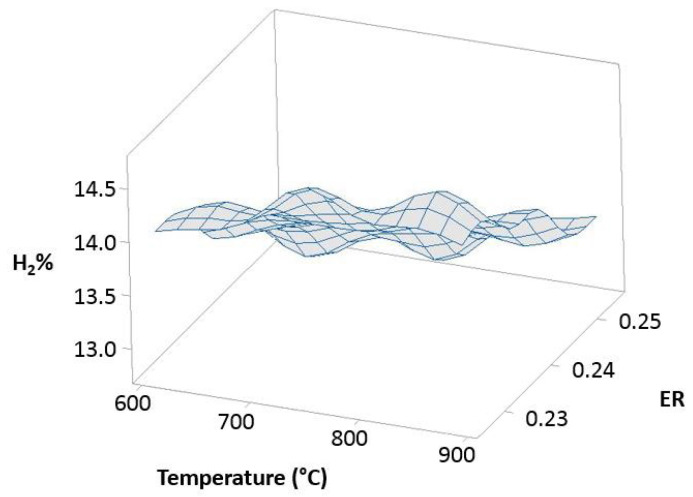
Exergetic efficiency of hydrogen concentration as a function of temperature and *ER* for 100% OLB.

**Figure 13 polymers-15-00750-f013:**
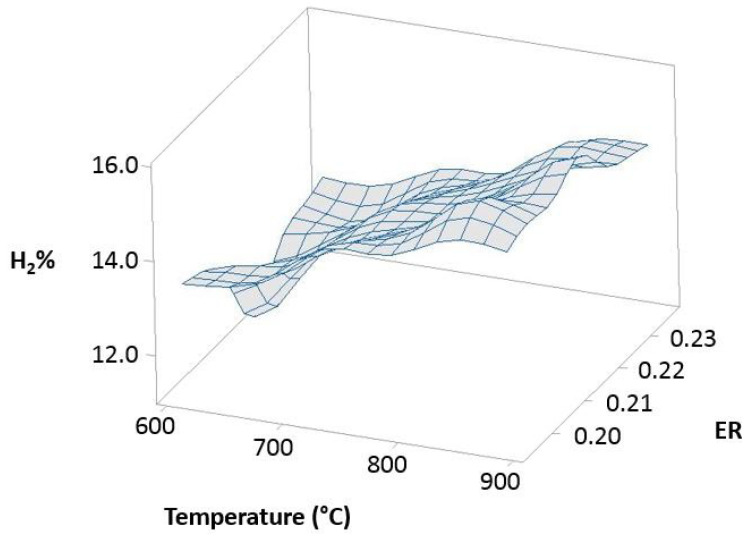
Exergetic efficiency of hydrogen concentration as a function of temperature and *ER* for the 10% 3DPPW.

**Figure 14 polymers-15-00750-f014:**
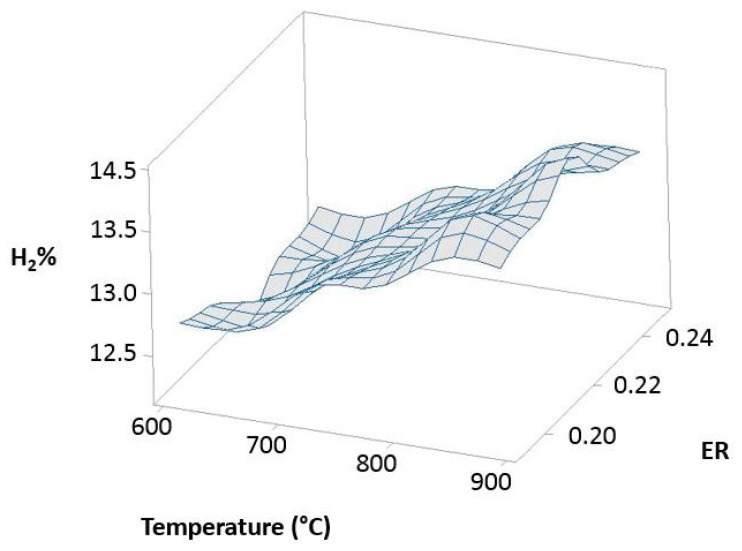
Exergetic efficiency of hydrogen concentration as a function of temperature and *ER* for the 20% 3DPPW.

**Table 1 polymers-15-00750-t001:** Fuel analysis.

	Parameters	Units	3DPPW	OLB
**Ultimate**	C	%	80.9	52.9
	H		13.4	7.7
	N		0.5	1.7
	S		1.9	0.1
	O		2.8	33.2
**Proximate**	Moisture	%	0.2	9.4
	Volatile		79.3	65.7
	Fixed Carbon		19.9	20.5
	Ashes		0.6	4.4
*HHV*		MJ/kg	40.7	20.3

**Table 2 polymers-15-00750-t002:** Gasification parameters.

		650 °C			750 °C			850 °C		
Parameters	Units	100% OLB	90% OLB 10% 3DPPW	80% OLB 20% 3DPPW	100% OLB	90% OLB 10% 3DPPW	80% OLB 20% 3DPPW	100% OLB	90% OLB 10% 3DPPW	80% OLB 20% 3DPPW
CO_2_	%	9.16	9.79	11.23	12.78	8.52	9.80	9.91	10.82	10.03
C_2_H_4_	%	0.17	0.58	0.37	0.82	0.47	0.21	0.56	0.63	0.42
C_2_H_6_	%	0.04	0.14	0.01	0.21	0.09	0.01	0.13	0.14	0.07
C_2_H_2_	%	0.04	0.03	0.01	0.02	0.02	0.01	0.03	0.25	0.02
H_2_S	%	0.05	0.05	0.05	0.05	0.05	0.05	0.05	0.05	0.31
N_2_	%	56.58	55.51	57.64	52.27	55.01	57.13	56.08	50.33	55.03
CH_4_	%	1.09	2.42	2.12	3.21	2.04	1.71	2.32	2.62	1.87
CO	%	20.92	19.34	18.59	16.11	19.91	20.11	18.31	19.44	20.69
H_2_	%	14.19	13.01	12.59	14.42	13.56	12.95	13.30	15.75	13.67
*LHV syngas*	MJ/Nm^3^	5.05	5.44	5.06	5.84	5.44	5.84	5.40	6.16	5.50
T rst	°C	658	662	652	748	756	699	720	708	697
T red	°C	397	492	368	505	562	476	515	547	484
P comb	KPa	−10	−16	−12	−14	−13	−12	−12	−10	−16
P React	KPa	−16	−22	−20	−19	−19	−27	−22	−25	−23
P filt	KPa	−26	−39	−34	−43	−35	−45	−40	−32	−46
Vair	m^3^/h	8.52	10.30	9.92	13.51	12.80	13.80	14.80	14.90	17.85
Tair	°C	13.50	18.70	12.00	12.30	19.10	14.40	15.80	19.60	14.90
Vtars	ml/h	146.80	127.96	111.54	134.41	119.41	108.32	151.21	137.73	138.94
Vchars	kg/h	0.16	0.12	0.13	0.14	0.11	0.12	0.16	0.16	0.18
*ER*	-	0.23	0.21	0.20	0.23	0.23	0.24	0.25	0.20	0.22
ṁ *syngas*	kg/h	8.72	15.94	14.29	24.67	20.85	20.64	23.18	30.90	29.31
ṁ *massic* *biomass*	kg/h	5.20	6.10	5.80	8.00	7.00	6.70	8.30	9.70	9.60
*LHV fuel*	MJ/Nm^3^	18.17	22.34	24.38	18.17	22.34	24.38	18.17	22.34	24.38
*CGE*	%	37.93	63.62	51.13	80.60	72.52	73.81	67.58	87.85	68.93
*ɳsyngas*	m^3^/kg	1.68	2.61	2.46	3.08	2.98	3.08	2.79	3.19	3.05
Exp. Time	min	360.00								

## Data Availability

We Accept MDPI Research Data Policies section at https://www.mdpi.com/ethics.

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
