# Peer review of "Energy Recovery from Polymeric 3D Printing Waste and Olive Pomace Mixtures via Thermal Gasification—Effect of Temperature"

_polymers, 2023, doi:10.3390/polym15030750_

Round 1

Reviewer 1 Report (Previous Reviewer 1)

1. The novelty of this study is really unclear. What is the reason of choosing these materials? The reviewer agrees that PETG has application in 3D printing field, although “2.3% of polymeric raw material used in additive manufacturing” is not significant. However, this cannot be a good reason to choose this material for this study.

2. This study is on gasification of olive pomace with incorporating low contents of PETG. However, the authors have highlighted the PETG as the main feedstock.

3. What is the influence of the additive manufacturing on the gasification process of these materials? The additive manufacturing has been highlighted in the manuscript; however, it affects neither the chemical composition nor the gasification process.

4. OLB abbreviation has been introduced in Line 26; however, it has been used in Line 24.

5. The authors claimed in the Introduction that “Poly (ethylene terephthalate) glycol (PETG) is a thermoplastic polyester commonly used in parts manufacturing through 3D printing technology.” What are the applications of PETG in the 3D printing technology? How much do these applications produce the waste? For instance, PETG is used in the biomedical applications in polymeric scaffolds fabrication; however, this application does not produce any waste.

6. In the last paragraph of the Introduction, the authors claimed that “there is a scarcity of pilot-scale studies with a specific focus on the application of multicriteria decision analysis techniques to select the most suitable temperatures for gasification” and “compare their performances using techniques of multicriteria decision making.” There is the multicriteria decision analysis neither in the methodology section nor in the results and discussion. It is noteworthy to mention that the application of multicriteria decision making method on the gasification process is not a new field and there are several publications in this field, for instance, https://doi.org/10.1016/j.fuel.2022.123362 and https://doi.org/10.1016/j.cep.2022.108961.

7. The literature survey on the gasification process is very narrow. It is recommended to expand this section.

8. The manuscript does not deserve the publication in its current format. It is strongly recommended to eliminate the subjects of 3D printing and additive manufacturing. The reviewer believes that these subjects are not related to the topic of the manuscript and presenting these subjects redundant. Also, it is strongly recommended not to highlight this manuscript as a gasification process of plastics. It is a biomass gasification by incorporating low contents of the plastic.

9. If the authors revise the manuscript based on these comments, the other sections need to be reviewed.

Author Response

Reviewer 1 report

Report structure:

*Reviewer consideration font

*Author response font

At first, this author appreciates and thanks time spent in this paper review. As well as the guidelines and corrections indicated that, from humility, helps me to improve in the field of research.

  1. The novelty of this study is really unclear. What is the reason of choosing these materials? The reviewer agrees that PETG has application in 3D printing field, although “2.3% of polymeric raw material used in additive manufacturing” is not significant. However, this cannot be a good reason to choose this material for this study.

- This manuscript study and evaluate the energy recovery of PETG polymer material through downdraft co-gasification using a combination of biomass and PETG as fuel of the thermochemical process. In section 1. Introduction mentioned datum of global utilization of PETG as 3D print material (2.3%) is used to contextualize the general state of this material as raw material in the industry of 3D print.The utilization of these materials in present research is due to two principal reasons:

  1. Authors previously have researched about this polymeric material, in lines 164-165, reference [31]*, a previous manuscript is cited. This manuscript is related with mechanical properties research of specimens and prototype parts which are manufactured through additive manufacturing technology using PETG as raw material. With this manuscript authors pretend the energy recovery of generated PETG residues. These 3D printed PETG residues are the raw material source used in the present manuscript for create the combined fuels of co-gasifications done.
  2. Olive pomace biomass is used due to tested results in other studies done [29, 30] demonstrate that in mixtures fuels to downdraft co-gasification with other polymeric materials work properly. Furthermore, geographical proximity of these resource is counted too in order to its choice and utilization to elaborate the energy recovery study of 3D printed manufactured PETG residues.

*[31] Mercado-Colmenero, J.M.; La Rubia, M.D.; Mata-Garcia, E.; Rodriguez-Santiago, M.; Martin-Doñate, C. Experimental and Numerical Analysis for the Mechanical Characterization of PETG Polymers Manufactured with FDM Technology under Pure Uniaxial Compression Stress States for Architectural Applications. Polymers 2020, 12, 2202. https://doi.org/10.3390/polym12102202.

**[29] Hermoso-Orzáez, M.J.; Mota-Panizio, R.; Carmo-Calado, L.; Brito, P. Thermochemical and Economic Analysis for Energy Recovery by the Gasification of WEEE Plastic Waste from the Disassembly of Large-Scale Outdoor Obsolete Luminaires by LEDs in the Alto Alentejo Region (Portugal). Appl. Sci. 2020, 10, 4601. https://doi.org/10.3390/app10134601.

**[30] Carmo-Calado, L.; Hermoso-Orzáez, M.J.; Mota-Panizio, R.; Guilherme-Garcia, B.; Brito, P. Co-Combustion of Waste Tires and Plastic-Rubber Wastes with Biomass Technical and Environmental Analysis. Sustainability 2020, 12, 1036. https://doi.org/10.3390/su12031036

  1. This study is on gasification of olive pomace with incorporating low contents of PETG. However, the authors have highlighted the PETG as the main feedstock.

-Statements have been corrected as requested by the reviewer:

Line 24: In the case of the study, co-gasification of olive pomace (OLB) was carried out with small additions of 3DPPW (10 and 20%) at different temperatures

  1. What is the influence of the additive manufacturing on the gasification process of these materials? The additive manufacturing has been highlighted in the manuscript; however, it affects neither the chemical composition nor the gasification process.

- The use of polymer residues allowed an increase in the concentration of hydrogen and carbon monoxide at high temperatures and at lower temperatures it capitalized on the formation of hydrocarbons, increasing the calorific value of the gas in most tests. These aspects are discussed in the results. Additive manufacturing is relevant in terms of necessary resource obtaining to elaborate mixed fuels. Author is agree with reviewer opinion, but, in consideration, it is necessary to express rigorously the scope of the manuscript, experimental tests are only have elaborated by using PETG residues coming from additive manufacturing parts.

  1. OLB abbreviation has been introduced in Line 26; however, it has been used in Line 24.

- The error has been fixed.

  1. The authors claimed in the Introduction that “Poly (ethylene terephthalate) glycol (PETG) is a thermoplastic polyester commonly used in parts manufacturing through 3D printing technology.” What are the applications of PETG in the 3D printing technology? How much do these applications produce the waste? For instance, PETG is used in the biomedical applications in polymeric scaffolds fabrication; however, this application does not produce any waste.

-In concordance with reviewer appreciation, other polymers more used in additive manufacturing parts exist, for example PLA. Authors understand that PETG characteristics are good enough for its utilization in additive manufacturing technology too. Raw material used in the elaboration of this manuscript are residues which are produced through 3D print parts, specimens and prototypes, made of PETG [31]*. According with the reviewer, currently applications of PETG in additive manufacturing are more limited than other polymers, however, PETG material use projection in biomedical industry as raw material to prosthesis elaboration become this material in a clear candidate in terms of global production increase. So it is necessary to search possible recovery solutions with a view to the near future for this material, anticipating its possible growth at the global level of production.

*[31] Mercado-Colmenero, J.M.; La Rubia, M.D.; Mata-Garcia, E.; Rodriguez-Santiago, M.; Martin-Doñate, C. Experimental and Numerical Analysis for the Mechanical Characterization of PETG Polymers Manufactured with FDM Technology under Pure Uniaxial Compression Stress States for Architectural Applications. Polymers 2020, 12, 2202. https://doi.org/10.3390/polym12102202.

  1. In the last paragraph of the Introduction, the authors claimed that “there is a scarcity of pilot-scale studies with a specific focus on the application of multicriteria decision analysis techniques to select the most suitable temperatures for gasification” and “compare their performances using techniques of multicriteria decision making.” There is the multicriteria decision analysis neither in the methodology section nor in the results and discussion. It is noteworthy to mention that the application of multicriteria decision making method on the gasification process is not a new field and there are several publications in this field, for instance, https://doi.org/10.1016/j.fuel.2022.123362 and https://doi.org/10.1016/j.cep.2022.108961.

- Dear reviewer, you are absolutely right, the sentence has been redone.

  1. The literature survey on the gasification process is very narrow. It is recommended to expand this section.

-The section was expanded. Speaking of the problems associated with the gasification of polymeric waste, addressing the benefits of mixing with biomass and addressing the different types of gasification reactors.

  1. The manuscript does not deserve the publication in its current format. It is strongly recommended to eliminate the subjects of 3D printing and additive manufacturing. The reviewer believes that these subjects are not related to the topic of the manuscript and presenting these subjects redundant. Also, it is strongly recommended not to highlight this manuscript as a gasification process of plastics. It is a biomass gasification by incorporating low contents of the plastic.

-According to the reviewer appreciation he is right that to name "gasification of PETG" is not rigorous due to experimental tests done are certainly a co-gasification of PETG and olive pomace biomass. However, 100% polymeric material gasification is not viable or functional under downdraft gasification boundary conditions because gasification device reactor clogging (lines 112-115) [30]*.  So, in this case, process cannot be automated in industrial way. Anyway, and in concordance with reviewer, there are other gasification technologies that allow the gasification of 100% polymers materials like plasma gasification method but are far of the scope of this manuscript. In introduction section is better marked that experimental tests carried out have been realized in terms of co-gasification agreeing reviewer opinion.

* [30] Carmo-Calado, L.; Hermoso-Orzáez, M.J.; Mota-Panizio, R.; Guilherme-Garcia, B.; Brito, P. Co-Combustion of Waste Tires and Plastic-Rubber Wastes with Biomass Technical and Environmental Analysis. Sustainability 2020, 12, 1036. https://doi.org/10.3390/su12031036

  1. If the authors revise the manuscript based on these comments, the other sections need to be reviewed.

-The authors thank the reviewer for his advice, corrections, considerations, appreciations and help, which help us to grow and improve within the field of research.

Reviewer 2 Report (Previous Reviewer 2)

The aim of the study was to investigate the possibility of energy recovery from plastic waste generated after the implementation of additive manufacturing projects, and then to apply a multipurpose optimization criterion to the co-gasification process using the methodology and response surface analyses, and to compare their performances using multi-criteria decision-making techniques.

The discussion of the results is in the form of a report, so please reword the text and give it a few concise headings not related to specific analyzes but to the issues being solved. The manuscript contains also a lot of mistakes (some highlighted below), which should be corrected.

Line 14-15, 560: “polyethylene terephthalate glycol”; Please use the IUPAC nomenclature.

Line 16 and 47: “Additive manufacturing”: It is not a proper name and it should not be capitalized. Please correct. Please correct throughout the text typographic errors such as “Waste to energy”, “Ratio” or “Higher heating value” written with a capital letter while the others are small or in the middle of sentence.

Line 19: “3D-printed polymeric waste”; The authors probably did not mean "to print the waste", and this is what this sentence implies. Rather "polymer waste from 3D printing". Please make a language correction.

Line 24-26: “OLB”; Please explain the abbreviations in the Abstract where they are first used. Please correct.

Line 33: “3Dprint waste”; There is no space. Please correct.

Table 1 should be in paragraph 3.1.

Author Response

Reviewer 2 report

Report structure:

*Reviewer consideration font

*Author response font

At first, this author appreciates and thanks time spent in this paper review. As well as the guidelines and corrections indicated that, from humility, helps me to improve in the field of research.

The aim of the study was to investigate the possibility of energy recovery from plastic waste generated after the implementation of additive manufacturing projects, and then to apply a multipurpose optimization criterion to the co-gasification process using the methodology and response surface analyses, and to compare their performances using multi-criteria decision-making techniques.

  1. The discussion of the results is in the form of a report, so please reword the text and give it a few concise headings not related to specific analyzes but to the issues being solved. The manuscript contains also a lot of mistakes (some highlighted below), which should be corrected.

-Restructuring of section 3. Results and discussion has been done according to the reviewer commentary, hope that the new format adaptation meet the expectations of the reviewer. Author thanks the constructive opinion of the reviewer.

  1. Line 14-15, 560: “polyethylene terephthalate glycol”; Please use the IUPAC nomenclature.

- According to the reviewer observation, IUPAC nomenclature was used to name the material as “poly (ethylene terephthalate) glycol” along the manuscript.

  1. Line 16 and 47: “Additive manufacturing”: It is not a proper name and it should not be capitalized. Please correct. Please correct throughout the text typographic errors such as “Waste to energy”, “Ratio” or “Higher heating value” written with a capital letter while the others are small or in the middle of sentence.

- Mistakes have been reviewed and fixed according reviewer advices.

  1. Line 19: “3D-printed polymeric waste”; The authors probably did not mean "to print the waste", and this is what this sentence implies. Rather "polymer waste from 3D printing". Please make a language correction.

- Reviewer appreciations have been fixed.

  1. Line 24-26: “OLB”; Please explain the abbreviations in the Abstract where they are first used. Please correct.

-Fixed error in line 24-26 as suggested by reviewer.

  1. Line 33: “3D print waste”; There is no space. Please correct.

- Correction has been done.

  1. Table 1 should be in paragraph 3.1.

- Table 1 was moved according to reviewer appreciation.

Round 2

Reviewer 1 Report (Previous Reviewer 1)

The authors addressed the previous comments; however, there are still some points should be addressed before final decision.

1. The acronym of 3DPPW has been used in line 19; however, it has not been defined.

2. The last paragraph of the Introduction should be modified considerably. The novelties and contributions of the study should be clearly highlighted in the last paragraph. Which gaps of the literature have been covered by this study?

3. In the last paragraph of the Introduction, it has been mentioned that “to apply a multipurpose optimization criterion to the co-gasification, using methodology and response surface analyses.” Do you sure that this optimization has been carried out in the study? There is no methodology description and results on the response surface methodology (RSM). RSM is a statistical optimization technique and there are no results of RSM. Only three surface plots have been presented in Figures 12-14 and there is no multi-objective optimization in the manuscript. RSM has many analyses such as analysis of variance, multi-objective optimization, regression models, and etc.

4. The previous comment #5 was “The authors claimed in the Introduction that “Poly (ethylene terephthalate) glycol (PETG) is a thermoplastic polyester commonly used in parts manufacturing through 3D printing technology.” What are the applications of PETG in the 3D printing technology? How much do these applications produce the waste? For instance, PETG is used in the biomedical applications in polymeric scaffolds fabrication; however, this application does not produce any waste.” The authors responded that PETG has applications in 3D printing. However, it is necessary to provide the information about the waste of PETG. As explained, many applications may produce no waste. It is necessary to address how much waste is produced in 3D printing of PETG to show the importance of energy recovery of PETG waste generated from 3D printing. They may not have any waste and their thermal treatment and energy recovery are not important. Without this information, the previous comment “It is strongly recommended to eliminate the subjects of 3D printing and additive manufacturing. The reviewer believes that these subjects are not related to the topic of the manuscript and presenting these subjects redundant.” is still valid.

5. Please present Figures 9-11 in white background and improve their quality.

6. Please present Figures 12-14 in white background and improve their quality.

Author Response

Reviewer 1 report

Report structure:

*Reviewer consideration font

*Author response font

At first, this author appreciates and thanks time spent in this paper review. As well as the guidelines and corrections indicated that, from humility, helps me to improve in the field of research.

  1. The acronym of 3DPPW has been used in line 19; however, it has not been defined.

- Reviewer suggestion has been corrected.  Line 19: “Polymer waste from 3D printing, hereinafter 3DPPW”

  1. The last paragraph of the Introduction should be modified considerably. The novelties and contributions of the study should be clearly highlighted in the last paragraph. Which gaps of the literature have been covered by this study?

- According to reviewer considerations, introduction section last paragraph has been modified.

  1. In the last paragraph of the Introduction, it has been mentioned that “to apply a multipurpose optimization criterion to the co-gasification, using methodology and response surface analyses.” Do you sure that this optimization has been carried out in the study? There is no methodology description and results on the response surface methodology (RSM). RSM is a statistical optimization technique and there are no results of RSM. Only three surface plots have been presented in Figures 12-14 and there is no multi-objective optimization in the manuscript. RSM has many analyses such as analysis of variance, multi-objective optimization, regression models, and etc.

- According to reviewer appreciations, introduction section last paragraph has been modified in order to clarify the aim and the scope of the research.

  1. The previous comment #5 was “The authors claimed in the Introduction that “Poly (ethylene terephthalate) glycol (PETG) is a thermoplastic polyester commonly used in parts manufacturing through 3D printing technology.” What are the applications of PETG in the 3D printing technology? How much do these applications produce the waste? For instance, PETG is used in the biomedical applications in polymeric scaffolds fabrication; however, this application does not produce any waste.” The authors responded that PETG has applications in 3D printing. However, it is necessary to provide the information about the waste of PETG. As explained, many applications may produce no waste. It is necessary to address how much waste is produced in 3D printing of PETG to show the importance of energy recovery of PETG waste generated from 3D printing. They may not have any waste and their thermal treatment and energy recovery are not important. Without this information, the previous comment “It is strongly recommended to eliminate the subjects of 3D printing and additive manufacturing. The reviewer believes that these subjects are not related to the topic of the manuscript and presenting these subjects redundant.” is still valid.

- Attending to reviewer demands, this ask can be answered through two points of view.  Locally, authors have already worked with this material previously [1] and understand that specific characteristics offered by PETG become it, in cases, a better choice to prototypes elaboration than others polymer, for example PLA, in the research field. FDA (food and drugs administration) [2] credit PETG as a good food container and become this material a fantastic option to elaborate prototypes of containers, bottles , glasses or cups related with  olive oil industry widely extended and important in the territory from authors belong. At global scale, marketers companies of 3D printing polymeric materials place PETG as one of favorite materials among customers due to its characteristics for 3D printing manufacture. Reports indicate PETG counts with 23% of market share situating this material in the second place only surpassed by PLA [3]. In the industrial field, PETG is widely commercialized, for example, in dental industry [4]. Transparent dental aligners are manufactured using additive manufacturing through CAD and inverse engineering processes and are provided as personalize service to customers [5, 6]. Dental aligners generally are changed gradually by another depending on facultative specific studies developed, the duration of each aligner reach 15 or 30 days usually. Dental aligners are manufactured using additive manufacturing technology and after their use do not account with another specific one, so parts become waste. Authors understand that this research, which experimental tests have made using exclusively PETG waste from 3D printing, can help to cushion possible environmental troubles related with dental industry and other industry fields [7] (where PETG is manufactured through additive manufacturing technology) through energy recovery of PETG, as described in the manuscript, anticipating its possible growth at the global level of production in the near future [8, 9].

*[1] Mercado-Colmenero, J.M.; La Rubia, M.D.; Mata-Garcia, E.; Rodriguez-Santiago, M.; Martin-Doñate, C. Experimental and Numerical Analysis for the Mechanical Characterization of PETG Polymers Manufactured with FDM Technology under Pure Uniaxial Compression Stress States for Architectural Applications. Polymers 2020, 12, 2202. https://doi.org/10.3390/polym12102202.

*[2] https://www.filamentive.com/product-category/rpetg-recycled-petg-pet-3d-printer-filament/

*[3] https://www.filamentive.com/2022-3d-printing-filament-trend-report/

*[4] http://products.scheu-dental.com/documents/5000/1-DOC/0/0/0/0/3/DURAN_BPZ_0146_GB_Original_3425.pdf

*[5]https://repositorio.autonoma.edu.co/bitstream/11182/82/1/Criterios_selecci%C3%B3n_material_base_dise%C3%B1o_termoformado.pdf

*[6]Álvarez, César. (2019). Comparison of mechanical properties of polymer sheets used in the manufacture of dental aligners. Link: https://www.researchgate.net/publication/336881917_Comparison_of_mechanical_properties_of_polymer_sheets_used_in_the_manufacture_of_dental_aligners

*[7] https://www.makeshaper.com/2020/05/28/various-industries-use-petg-filament-for-3d-printing/

*[8] https://idataresearch.com/product/dental-3d-printing-market/

*[9] https://dataintelo.com/report/petg-market/

  1. Please present Figures 9-11 in white background and improve their quality.

- Reviewer suggestions have been improve.

  1. Please present Figures 12-14 in white background and improve their quality.

- Reviewer suggestions have been improve.

Reviewer 2 Report (Previous Reviewer 2)

Dear Authors,

"Energy recovery from polymeric 3D-printed polymeric waste"; In the title and line 585 still "waste printed in 3D" meaning and why two times polymeric in the title!

Line 108, 4403-405, 471; Why do authors use capital letters in the middle of a sentence?

Line 14, 40, 565; There should be no space after poly in the name.

Author Response

Reviewer 2 report

Report structure:

*Reviewer consideration font

*Author response font

At first, this author appreciates and thanks time spent in this paper review. As well as the guidelines and corrections indicated that, from humility, helps me to improve in the field of research.

  1. "Energy recovery from polymeric 3D-printed polymeric waste"; In the title and line 585 still "waste printed in 3D" meaning and why two times polymeric in the title!

- Line 585 was rewritten to a better comprehension attending reviewer advice. Corrected word repetition in the title too.

  1. Line 108, 4403-405, 471; Why do authors use capital letters in the middle of a sentence?

- Suggested corrections have been implicated according reviewer appreciations.

  1. Line 14, 40, 565; There should be no space after poly in the name.

- Corrections have been done according reviewer appreciations.

Round 3

Reviewer 1 Report (Previous Reviewer 1)

The authors have addressed the comments and the revisions seem to be desirable; however, it is recommended to provide some of the explanations provided for the comment #4 in the manuscript before the final acceptance.

Author Response

Reviewer 1 report

Report structure:

*Reviewer consideration font

*Author response font

At first, this author appreciates and thanks time spent in this paper review. As well as the guidelines and corrections indicated that, from humility, helps me to improve in the field of research.

  1. The authors have addressed the comments and the revisions seem to be desirable; however, it is recommended to provide some of the explanations provided for the comment #4 in the manuscript before the final acceptance.

- According reviewer suggestions, solicited content, related with PETG uses as 3D printing material in industrial environment, has been added to the manuscript. Lines 49-61. Related bibliography references where the new text is based have been aggregated too. [8-12]

This manuscript is a resubmission of an earlier submission. The following is a list of the peer review reports and author responses from that submission.

Round 1

Reviewer 1 Report

This manuscript studies the gasification process of biomass by incorporating the PETG. The novelty of the study is not clear and the manuscript does not deserve the publication in this format.

1. What is the novelty of this study? As you know, the gasification process is affected by the chemical composition of the feedstock and the processing conditions. The FDM process does not affect the chemical composition. So, what is the purpose of using the PETG waste of the FDM process. In this case, there is not any difference between the PETG of the FDM process and the PETG of the injection process, for instance.

2. The authors specified the introduction, the title, and the literature to the FDM process. However, the main subject of the study is biomass gasification by incorporating the PETG.

3. The title has serious issue. The gasification has been done for biomass and PETG is only an additive with 10 and 20 wt%. However, the title shows that the gasification has been done for PETG.

4. Please avoid using the abbreviations in the title.

5. Please shorten the abstract length.

6. The literature survey needs a major revision. The gasification process of plastic waste should be reviewed.

7. OLB has been used in Line 108; however, it has been introduced in Line 114.

8. Line 526 “DOE (design of Experiments) software”, DOE is not a software. It is a technique. The results indicate that the authors have used the Minitab software.

9.Please provide the figures with higher qualities especially Figures 9-14.

Author Response

Reviewer 1 report

Report structure:

*Reviewer consideration font

*Author response font

At first, this author appreciates and thanks time spent in this paper review. As well as the guidelines and corrections indicated that, from humility, helps me to improve in the field of research.

  1. What is the novelty of this study? As you know, the gasification process is affected by the chemical composition of the feedstock and the processing conditions. The FDM process does not affect the chemical composition. So, what is the purpose of using the PETG waste of the FDM process. In this case, there is not any difference between the PETG of the FDM process and the PETG of the injection process, for instance.

-Manuscript starts with a raw material, in this case PETG, which has already been manufactured using FDM technology. From there, experimental tests are carried out and, finally, paper ends with a statistical analysis. This allows the interpolation of the data looking for the optimization of the process to predict the proportion of PETG within the fuel and temperature necessary for an optimal energy recovery of 3D printing waste material, thus avoiding carrying out more experimental tests that are expensive and require preparation time. With reference to the manuscript, the text has been modified to make the novelty more detectable to the reader in accordance with the reviewer's proposal.

  1. The authors specified the introduction, the title, and the literature to the FDM process. However, the main subject of the study is biomass gasification by incorporating the PETG.

-Title has been modified to better express the aim and scope of the article in accordance with the advice of the reviewer, the abstract, introduction and part of the literature have also been rewritten.

  1. The title has serious issue. The gasification has been done for biomass and PETG is only an additive with 10 and 20 wt%.However, the title shows that the gasification has been done for PETG.

- Title has been modified to better express the aim and scope of the research in accordance with the advice of the reviewer.

  1. Please avoid using the abbreviations in the title.

-Title has been changed as suggested by reviewer

  1. Please shorten the abstract length.

-The abstract has been written to better express the aim and scope of the research as suggested by the reviewer.

  1. OLB has been used in Line 108; however, it has been introduced in Line 114.

-According to the reviewer, the consideration has been modified.

  1. Line 526 “DOE (design of Experiments) software”, DOE is not a software. It is a technique. The results indicate that the authors have used the Minitab software.

-According to the reviewer, the consideration has been modified.

  1. Please provide the figures with higher qualities especially Figures 9-14.

-According to the reviewer, the quality of the mentioned figures has been increased.

Reviewer 2 Report

The aim of the study was to investigate the possibility of energy recovery from plastic waste generated after the implementation of additive manufacturing projects. The process of recovering energy from these materials was carried out by gasification of olive pit biomass. The biomass was mixture with by-products obtained from various studies to characterize poly(ethylene terephthalate) glycol. The sources of polymer material used in this study are waste from 3D printed specimens used in mechanical testing and parts created for 3D printing, which were scale models also designed for experimental mechanical tests.

The aim of the work is marked but gets lost in the extensive description. Also there is no clearly presented novelty for the Readers. Please include clear aim and explicit novelty to the abstract. The discussion of the results is in the form of a report, which also makes it difficult to understand the topic, so please reword the text and give it a few concise headings not related to specific analyzes but to the issues being solved. The manuscript contains also a lot of mistakes (some highlighted below), which should be corrected.

1. Two sentences in the title should be separated by a dash or colon. Please correct.

2. Line 5: Was the author removed on purpose or by accident? Please correct.

3. Line 20, 57 and 91: “4.0 industry”; I'm not on the subject of the next industrial revolution, but Industry 4.0 is probably a proper name written with a capital letter and in a different order. Please correct.

4. Line 21, 101 and 569: “Polyethylene terephthalate glycol” Please use the IUPAC name of the polymers, also according to the IUPAC nomenclature, the names of polymers whose monomers consist of two words or more are written with parentheses. In addition, words written in the middle of a sentence that are not a proper name are written with a lower case letter. Please correct.

5. Line 29: “3DPPW”; Abbreviations should be limited in the abstract, especially since they are neither explained nor repeated. Please correct.

6. Line 3: “3Dprint waste”; There is no space. Please correct.

7. Line 108 and 114: Abbreviations should be explained where they are first used. Please correct.

8. Line 129, 231: Words written in the middle of a sentence that are not a proper name are written with a lower case letter. Please correct.

9. Line 129: There is no abbreviation for sulfur. If the authors explain all of them then one should not be omitted.

10. Line 166: “carbon (C)” No point in explaining it again.

11. From line 187: According to the SI, variables should be in italics (e.g. F). Please correct throughout the text.

12. Line 198: “0, 2 and 0, 4”; Is it 0 and 2 or 0.2? Please correct.

Author Response

Report structure:

*Reviewer consideration font

*Author response font

At first, this author appreciates and thanks time spent in this paper review. As well as the guidelines and corrections indicated that, from humility, helps me to improve in the field of research.

The aim of the study was to investigate the possibility of energy recovery from plastic waste generated after the implementation of additive manufacturing projects. The process of recovering energy from these materials was carried out by gasification of olive pit biomass. The biomass was mixture with by-products obtained from various studies to characterize poly (ethylene terephthalate) glycol. The sources of polymer material used in this study are waste from 3D printed specimens used in mechanical testing and parts created for 3D printing, which were scale models also designed for experimental mechanical tests.

The aim of the work is marked but gets lost in the extensive description. Also there is no clearly presented novelty for the Readers. Please include clear aim and explicit novelty to the abstract. The discussion of the results is in the form of a report, which also makes it difficult to understand the topic, so please reword the text and give it a few concise headings not related to specific analyzes but to the issues being solved. The manuscript contains also a lot of mistakes (some highlighted below), which should be corrected.

The paper has been partially modified trying to show greater linearity in text and better aim recognition, as well as the novelty of the described procedure for the energy recovery of this material (PETG) used for 3D printing. Currently there is no report in scientific literature that faces this specific issue about this kind of waste recovery. Also including a numerical model that allows extrapolating the experimental results obtained at different temperatures without the need to repeat the gasification processes in the case of this material.

  1. Two sentences in the title should be separated by a dash or colon. Please correct

-Paper title has been modified to better express of research aim, in accordance with reviewer advices.

  1. Line 5: Was the author removed on purpose or by accident? Please correct.

-Corrected introduction lines from 36-87, in consideration of reviewer advice to better expose of the aim of the research.

  1. Line 20, 57 and 91: “4.0 industry”; I'm not on the subject of the next industrial revolution, but Industry 4.0 is probably proper name written with a capital letter and in a different order. Please correct.

-Same as previous.

  1. Line 21, 101 and 569: “Polyethylene terephthalate glycol “Please use the IUPAC name of the polymers, also according to the IUPAC nomenclature, the names of polymers whose monomers consist of two words or more are written with parentheses. In addition, words written in the middle of sentence that are not a proper name are written with a lowercase letter. Please correct.

-Fixed error in line 569 as suggested by reviewer.

  1. Line 29: “3DPPW”; Abbreviations should be limited in the abstract, especially since they are neither explained nor repeated. Please correct.

-Abbreviations introduction has been rewrote following reviewer proposal.

  1. Line 3: “3Dprint waste”; There is no space. Please correct

-Line 3 has been rewrote to better overall understanding of text, including aim and novelty of the research in consideration of reviewer advices.

  1. Line 108 and 114: Abbreviations should be explained where they are first used. Please correct.

-Abbreviations have been correctly defined in the new text.

  1. Line 129, 231: Words written in the middle of a sentence that are not a proper name are written with a lower case letter. Please correct.

-According to the reviewer, cited mistake have been fixed.

  1. Line 129: There is no abbreviation for sulfur. If the authors explain all of them then one should not be omitted.

-According to the reviewer, the consistency errors in the new text have been fixed.

  1. Line 166: “carbon (C)” No point in explaining it again.

-According to the reviewer, the consistency errors in the new text have been fixed.

  1. From line 187: According to the SI, variables should be in italics (e.g. F). Please correct throughout the text.

-According to the reviewer, the consideration has been modified

  1. Line 198: “0, 2 and 0, 4”; is it 0 and 2 or 0.2? Please correct.

-Numerical error has been corrected.